# Improving On-policy Learning with Statistical Reward Accumulation

## Abstract

Deep reinforcement learning has obtained significant breakthroughs in recent years. Most methods in deep-RL achieve good results via the maximization of the reward signal provided by the environment, typically in the form of discounted cumulative returns. Such reward signals represent the immediate feedback of a particular action performed by an agent. However, tasks with sparse reward signals are still challenging to on-policy methods. In this paper, we introduce an effective characterization of past reward statistics (which can be seen as long-term feedback signals) to supplement this immediate reward feedback. In particular, value functions are learned with multi-critics supervision, enabling complex value functions to be more easily approximated in on-policy learning, even when the reward signals are sparse. We also introduce a novel exploration mechanism called "hot-wiring" that can give a boost to seemingly trapped agents. We demonstrate the effectiveness of our advantage actor multi-critic (A2MC) method across the discrete domains in Atari games as well as continuous domains in the MuJoCo environments. A video demo is provided at `https://youtu.be/zBmpf3Yz8tc` and source codes will be made available upon paper acceptance.

## 1 Introduction

Advances in deep learning have mobilized the research community to adopt deep reinforcement learning (RL) agents for challenging control problems, typically in complex environments with raw sensory state-spaces. Breakthroughs by Mnih et al. (2015) show RL-agents can reach above-human performance in Atari 2600 games, and AlphaGo Zero Silver et al. (2017) becomes the world champions on the game of *Go*. Still, training RL agents is non-trivial. Off-policy methods typically require days of training to obtain competitive performance, while on-policy methods could be trapped in local minima. Recent techniques featuring on-policy learning Mnih et al. (2016); Schulman et al. (2017); Wu et al. (2017) have shown promising results in stabilizing the learning processes, enabling an agent to solve a variety of tasks in much less time. In particular, the state-of-the-art on-policy ACKTR agent by Wu et al. (2017) shows improved sample efficiency with the help of Kronecker-factored (K-Fac) approximate curvature for natural gradient updates, resulting in stable and effective model updates towards a more promising direction.

However, tasks with sparse rewards remain challenging to on-policy methods. An agent could take massive amount of exploration before reaching non-zero rewards; and as the agent learns on-policy, the sparseness of reward feedback (receiving all-zero rewards from most actions performed by the agent) could be malicious and render an agent to falsely predict all states in an environment towards a value of zero. As there does not exist a universal criterion for measuring "task sparseness", we show an ad-hoc metric in Figure 1 in an attempt to provide intuition. For example, we observe that the ACKTR agent is unable to receive sufficient non-zero immediate rewards that can provide instructive agent updates in Atari games "Freeway" and "Enduro", resulting in failures when solving these two games. Moreover, if ACKTR gets drawn to and trapped in unfavorable states (as in games like Boxing and WizardOfWor), it could again take long hours of exploration before the agent can get out of the local minima. Such evidence shows that on-policy agent could indeed suffer from the insufficiencies of guidance provided by the exclusive immediate reward signals from the environment.

In this paper, we introduce an effective auxiliary reward signal in tasks with sparse rewards to remedy the deficiencies of learning purely from standard immediate reward feedbacks. As on-policy

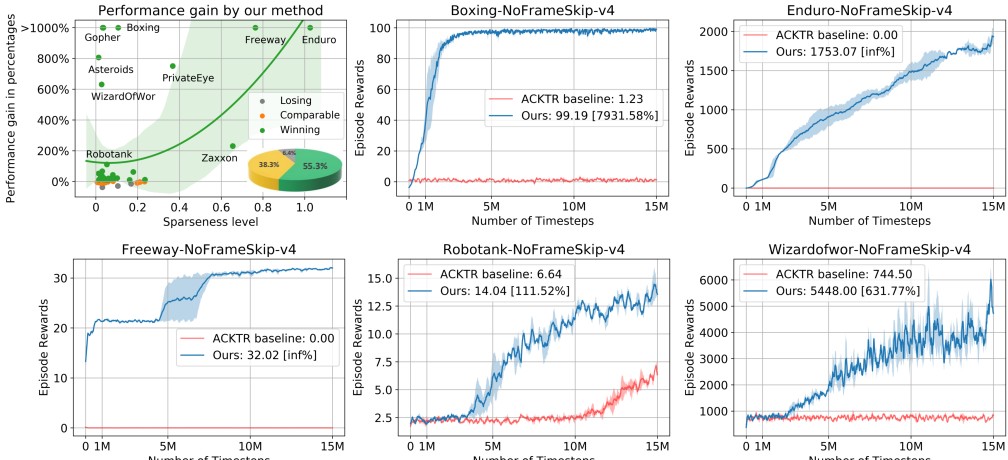

Figure 1: Performance of A2MC on Atari games trained with 15 million timesteps. Our method has a winning rate of $55.3\%$ among all the Atari games tested, as compared to the ACKTR. Our A2MC learns quickly in some of the hardest games for on-policy methods, such as "Boxing", "Enduro", "Freeway", "Robotank" and "WizardOfWor". The sparseness of a game is defined as the sparseness of average rewards $\mathbf{x}$ obtained by ACKTR within the first $n = 10^6$ timesteps by $\varphi(\mathbf{x}) = \left(\sqrt{n} - \frac{\|\mathbf{x}\|_1}{\|\mathbf{x}\|_2}\right) / (\sqrt{n} - 1)$. A higher value of sparseness indicates sparser rewards. A relative performance margin (in terms of final reward) larger than $10\%$ is deemed as winning / losing. The shaded region denotes the standard deviation over 2 random seeds.

agents may take many explorations before reaching non-zero immediate rewards, we argue that we can leverage past reward statistics to provide more instructive feedbacks to agents in the same environment. To this end, we propose to characterize the past reward statistics in order to gauge the "long-term" performance of an agent (detailed in Section 4). After performing an action, an agent will receive a long-term reward signal describing its past performance upon this step, as well as the conventional immediate reward from the environment. To effectively characterize the long-term performance of the agent, we take into considerations both the crude amount of rewards and the volatility of rewards received in the past, where highly volatile distributions of long-term rewards are explicitly penalized. This enables complex value functions to be more easily approximated in multi-critics supervision. We find in practice that by explicitly penalizing highly volatile long-term rewards while maximizing the expectation of short-term rewards, the learning process and the overall performance are improved regarding both sample efficiency and final rewards. We further propose a "hot-wiring" exploration mechanism that can boost seemingly trapped agent in the earlier stage of learning. By leveraging the characterization of long/short-term reward statistics, our proposed advantage actor multi-critic model (A2MC) shows significantly improved performance on the Atari 2600 games and the MuJoCo tasks as compared to the state-of-the-art on-policy methods.

## 2 RELATED WORK

**Reward shaping and pseudo-rewards**: To tackle the challenge in tasks with rarely observed rewards, pseudo-rewards maximization is adopted in earlier works Konidaris & Barto (2009); Silver & Ciosek (2012). Auxiliary vision tasks (e.g., learning pixel changes or network features) are adopted in the off-policy *UNREAL* agent Jaderberg et al. (2016) in order to facilitate learning better feature representations, particularly for sparse reward environments. Another direction of effort involves directly engineering a better reward function or shaping the reward signal. Andrychowicz et al. (2017) enhances off-policy learning by re-producing informative reward in hindsight for sequences of actions that do not lead to success previously. The HRA approach Van Seijen et al. (2017) exploits domain knowledge to define a set of environment-specific rewards based on reward categories. And the winning approach that learns playing "Doom" Lample & Chaplot (2017) shows promising success in the FPS game that carefully crafting the task rewards would indeed be beneficial. In contrast

to heuristically defining vision-related auxiliary tasks, our proposed A2MC agent learns from the characterization of intrinsic past reward statistics obtainable from any environment; and different from the hybrid architecture pertaining to Ms. Pacman only and the reward shaping settings tailored specifically to "Doom", our proposed reward characterization mechanism is generic and our A2MC generalizes well to a variety of tasks without the need to engineer a decomposition of problem-specific environment rewards. Moreover, the capability of the proposed method to further boost reward shaping is evidenced in our case study on playing Doom (see Appendix F).

**Multi-agents:** The *multi-agent* approaches Lanctot et al. (2017); Lowe et al. (2017); Jin et al. (2018) present another promising direction for learning. They propose to train multiple agents in parallel when solving a task, and to combine multiple action-value functions with a centralized action-value function. The multi-critics supervision in our proposed A2MC model can be seen as a form of joint-task or multi-task learning Teh et al. (2017) for both long-term and short-term rewards.

**On-policy v.s. Off-policy:** Our empirical results based on learning the characterization of long/short-term reward statistics also echo the effectiveness of a recently proposed off-policy reinforcement learning framework Bellemare et al. (2017) that features a distributional variant of Q-learning, wherein the value functions are learned to match the distribution of standard immediate returns. Also, Wang et al. (2016) shows that applying experience replay to on-policy methods can further enhance learning stability. Schulman et al. (2016) proposes a variant of advantage function using *eligibility traces* that provides both low-variance and low-bias gradient estimates. These works are orthogonal to our approach can potentially be combined with the proposed characterization of past reward statistics to further enhance learning performance. While our extensive experiments (see also Appendix E and Appendix F) show promising results of our approach in both on- and off-policy frameworks, we focus on "on-policy" methods (i.e., those that do not involve off-policy trajectories or experience replay) as in Wu et al. (2017) in the main text in order to systematically evaluate the potential of our proposed reward mechanism within the scope of this work.

## 3 PRELIMINARY

Consider the standard reinforcement learning setting where an agent interacts with an environment over a number of discrete time step. At each time step $t$, the agent receives an environment state $s_t$, then executes an action $a_t$ based on policy $\pi_t$. The environment produces reward $r_t$ and next state $s_{t+1}$, according to which the agent gets feedback of its immediate action and will decide its next action $a_{t+1}$. The process $< \mathbf{S}, \mathbf{A}, \mathbf{R}, \mathbf{S} >$, typically considered as a Markov Decision Process, continues until a terminal state $s_T$ upon which the environment resets itself and produces a new episode. Under conventional settings, the return is calculated as the discounted summation of rewards $r_t$ accumulated from time step $t$ onwards $R_t = \sum_{k=0}^{\infty} \gamma^k r_{t+k}$. The goal of the agent is to maximize the expected return from each state $s_t$ while following policy $\pi$. Each policy $\pi$ has a corresponding action-value function defined as $Q^\pi(s, a) = \mathbb{E}[R_t | s_t = s, a_t = a; \pi]$. Similarly, each state $s \in S$ under policy $\pi$ has a value function defined as: $V^\pi(s) = \mathbb{E}[R_t | s_t = s]$. In **value-based approaches** (e.g., Q-learning Mnih et al. (2015)), function approximator $Q(s, a; \theta)$ can be used to approximate the optimal action value function $Q^*(s, a)$. This is conventionally learned by iteratively minimizing the below loss function:

$$L(\theta) = \mathbb{E}[(y_t^{target} - Q(s_t, a_t; \theta))^2], \tag{1}$$

where $y_t^{target} = r_t + \gamma \max_{a'} Q(s_{t+1}, a'; \theta)$ and $s_{t+1}$ is the next state following state $s_t$.

In **policy-based approaches** (e.g., policy gradient methods), the optimal policy $\pi^*(a|s)$ is approximated using the approximator $\pi(a|s; \theta)$. The policy approximator is then learned by gradient ascent on $\nabla_\theta \mathbb{E}[R_t] \approx \nabla_\theta \log \pi(a_t|s_t; \theta) R_t$. The REINFORCE method Williams (1992) further incorporates a baseline $b(s_t)$ to reduce the variance of the gradient estimator: $\nabla_\theta \mathbb{E}[R_t]_{REINFORCE} \approx \nabla_\theta \log \pi(a_t|s_t; \theta)(R_t - b(s_t))$

In **actor-critic based approaches**, the variance reduction further evolves into the advantage function $A(s_t, a_t) = Q(s_t, a_t) - V(s_t)$ in Mnih et al. (2016), where the action value $Q^\pi(s_t, a_t)$ is approximated by $R_t$ and $b(s_t)$ is replaced by $V^\pi(s_t)$, deriving the advantage actor-critic architecture where actor-head $\pi(\cdot|s)$ and the critic-head $V(s)$ share low-level features. The gradient update rule w.r.t. the action-head is $\nabla_\theta \log \pi(a_t|s_t; \theta)(R_t - V(s_t; \theta))$. The gradient update w.r.t. the critic-head is: $\nabla_\theta (R_t - V(s_t; \theta))^2$, where $R_t = r_t + \gamma V(s_{t+1})$.

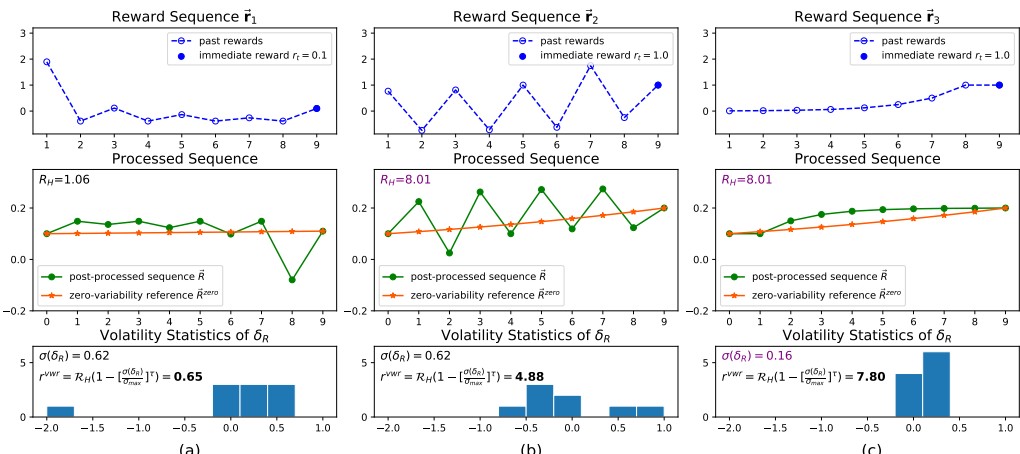

Figure 2: Illustration of the proposed variability-weighted reward (VWR). The first row shows the raw reward sequence (blue) while the second row presents the post-processed sequence $\vec{\mathcal{R}}$ (green) and the zero-variability reference $\vec{\mathcal{R}}^{zero}$ (orange), and $\mathcal{R}_H$ is calculated as a reflection of *how high the immediate reward is*. The third row shows the volatility statistics of $\delta_{\mathcal{R}}$, representing *how varied past rewards were*. We curated 3 hypothetical reward sequences – (a) highly varied sequence with low immediate reward, resulting in the lowest VWR; (b) highly varied sequence with high immediate reward, leading to a relatively high VWR; (c) stable sequence with high immediate reward, achieving the best VWR. More examples can be found in the Appendix A.

## 4   CHARACTERIZATION OF PAST REWARD STATISTICS

The conventional reward $r_t$ received from the environment at time step $t$ after an action $a_t$ is performed represents the immediate reward regarding this particular action. This "immediacy" could be interpreted as a *short-term* horizon of how the agent is doing, *i.e.,* evaluating the agent via judging its actions by immediate rewards. We argue that the deficiencies of learning solely from immediate rewards mainly come from this limitation that the agent is learning from one single type of exclusive short-term feedback.

As the goal of providing reward feedback to an agent is to inform the agent of its performance, we seek to find an auxiliary performance metric that can measure whether the agent is performing *consistently* well. Inspired by the formulation of *Sharpe Ratio* ($\mathbb{E}[r] \times \frac{1}{\sigma_r}$) in evaluating the long-term performance of porfolio strategies where the return $\mathbb{E}[r]$ is inversely weighted by the risk $\sigma_r$, an effective characterization of historical reward statistics should take into account at least two factors, namely 1) how high the immediate reward is and 2) how varied past rewards were, bringing the desired notion of "risk-adjusted return" as in Sharpe (1994).

### 4.1   VARIABILITY-WEIGHTED REWARD

To this end, we follow insights behind Dowd (2000); Sharpe (1994) and define a variability-weighted characterization of past rewards. This is illustrated in Figure 2. We consider a historical sequence of $T$ rewards upon timestep $t$ (looking backward $T-1$ timesteps): $\vec{\mathbf{r}} = [r_{t-(T-1)}..., r_{t-2}, r_{t-1}, r_t]$. In order to evaluate how high and varied the reward sequence is, a few steps of pre-processing $\mathcal{G}$ is applied, denoted as $\vec{\mathcal{R}} = \mathcal{G}(\vec{\mathbf{r}})$. Specifically, we first derive the reward change at each timestep (similar to the "differential return" concept in Sharpe (1994)) with $d_n = r_n - r_{n-1}$:

$$\vec{\mathbf{d}} = [d_{t-(T-1)}, d_{t-(T-2)}, \ldots, d_t] = [r_{t-(T-1)}, r_{t-(T-2)} - r_{t-(T-1)}, \ldots, r_t - r_{t-1}]. \quad (2)$$

Then we re-order the sequence by flipping [1] with $f_n = d_{t+1-n}$:

$$\vec{\mathbf{f}} = [f_1, f_2, \ldots, f_T] = [d_t, d_{t-1}, \ldots, d_{t-(T-1)}]. \quad (3)$$

---

[1]By flipping, we further encourage "recent" stable rewards and penalize the volatility of recent past rewards. A concrete example is given in the Appendix A.

Next we append $f_0 = 1$ to the head of sequence $\vec{\mathbf{f}}$ and take the normalized cumulative sum to obtain the post-processed reward sequence as $\vec{\mathcal{R}} = [\mathcal{R}_0, \mathcal{R}_1, \ldots, \mathcal{R}_T] = \frac{1}{T+1}[f_0, f_0 + f_1, \ldots, \sum_{i=0}^{T} f_i]$. Under such processing, numerical instability (see Eq. 4) when all rewards in the sequence are zero can be alleviated, while the averaging term $\frac{1}{T+1}$ mitigates the effect of introducing the artificial $f_0$.

The resulting $\vec{\mathcal{R}}$ is a reward sequence with $\mathcal{R}_T - \mathcal{R}_0 = \frac{1}{T+1} r_t$, and $\mathcal{R}_n - \mathcal{R}_{n-1} = \frac{1}{T+1}(r_{t+1-n} - r_{t-n})$. Therefore, the difference between $\mathcal{R}_T$ and $\mathcal{R}_0$ represents the immediate reward and the whole sequence $\vec{\mathcal{R}}$ reflects the volatility of past rewards. In Figure 2, three examples of processed sequence are presented in the second row with the corresponding raw rewards shown in the first row. We account for *how high the immediate reward is* by defining the relative percentage log total return as:

$$\mathcal{R}_H = 100 \times (e^{\frac{1}{T} \ln \frac{\mathcal{R}_T}{\mathcal{R}_0}} - 1) = \frac{\mathcal{R}_T^{1/T} - \mathcal{R}_0^{1/T}}{\mathcal{R}_0^{1/T}} \times 100. \tag{4}$$

To account for *how varied past rewards were*, we first define a smooth *zero-variability reference* as: $\vec{\mathcal{R}}^{zero} = [\mathcal{R}_0^{zero}, \mathcal{R}_1^{zero}, \ldots, \mathcal{R}_T^{zero}] = \mathcal{R}_0[e^{0 \times \widetilde{\mathcal{R}}}, e^{1 \times \widetilde{\mathcal{R}}}, \ldots, e^{T\widetilde{\mathcal{R}}}]$ with $\widetilde{\mathcal{R}} = \frac{1}{T} \ln \frac{\mathcal{R}_T}{\mathcal{R}_0}$, representing a smooth monotonic reference sequence from $\mathcal{R}_0$ to $\mathcal{R}_T$. Then we define the reward differential $\delta_\mathcal{R}$ as the differential reward versus its zero-variability reference as $\delta_\mathcal{R}(n) = \frac{\mathcal{R}_n - \mathcal{R}_n^{zero}}{\mathcal{R}_n^{zero}}$, whose statistics are sketched in the third row of Figure 2. With maximally allowed volatility as $\sigma_{max}$, the variability weights can be defined as: $\omega = 1 - [\frac{\sigma(\delta_\mathcal{R})}{\sigma_{max}}]^\tau$, where $\sigma(\cdot)$ is the standard deviation and $\tau$ controls the rate to penalize highly volatile reward distribution. Finally we can derive the variability-weighted past reward indicator $r^{vwr}$ for the characterization of past reward statistics:

$$r^{vwr} = \begin{cases} \mathcal{R}_H(1 - [\frac{\sigma(\delta_\mathcal{R})}{\sigma_{max}}]^\tau) & \text{if } \sigma(\delta_\mathcal{R}) < \sigma_{max}, \mathcal{R}_T > 0 \\ 0 & \text{otherwise} \end{cases} \tag{5}$$

The formulation of Equation 5 share principled themes as in Sharpe (1994) and Dowd (2000):

1. Dowd (2000) compares the newly obtained $\text{SR}^{new}$ with the previous $\text{SR}^{old}$ in choosing new assets; we derive $\mathcal{R}_H$ in Eq. 4 by comparing the latest reward $\mathcal{R}_T$ with $\mathcal{R}_0$ to explicitly encourage the agent to aim for reward improvements in "choosing new actions";
2. Both the Sharpe Ratio (SR) and Eq. 5 involve "variability weights" to adjust for the unit risk of return $\mathbb{E}[\mathcal{R}]$ Sharpe (1994) (i.e., $\frac{1}{\sigma_r}$ for SR and $1 - [\frac{\sigma(\delta_\mathcal{R})}{\sigma_{max}}]^\tau$ for $r^{vwr}$);
3. Whereas Dowd (2000) introduces the concept of "minimum required return" based on the elasticity of value at risk (VaR), we consider the maximum tolerance level $\sigma_{max}$ with elasticity controlled by $\tau$ for improved learning stability of $r^{vwr}$ (see also Appendix H).

Example computed values of $r^{vwr}$ for the characterization of different reward statistics are shown in Figure 2 and we show strong empirical results (in Section 6) to confirm the validity and robustness of the proposed formulation in multiple reinforcement learning domains.

## 4.2 MULTI-CRITIC ARCHITECTURE

A higher value of $r^{vwr}$ indicates better agent performance as the result of the historical sequence of actions. The same set of optimization procedures for conventional value function (*i.e.,* via maximization of immediate reward signal $r$) update can be applied accordingly. The actual returns computed from both the "long-term" and "short-term" rewards are discounted by the same factor $\gamma$. In particular, for standard $N$-step look-ahead approaches, we have:

$$R_t^{\text{short-term}} = \sum_{n=0}^{N-1} \gamma^n r_{t+n} + \gamma^N V(s_{t+N}), \quad R_t^{\text{long-term}} = \sum_{n=0}^{N-1} \gamma^n r_{t+n}^{vwr} + \gamma^N V^{vwr}(s_{t+N}) \tag{6}$$

Similar to the standard state value function $V(s)$, we further define $V^{vwr}(s)$ as the value function w.r.t the variability-weighted reward $r^{vwr}$. These value functions form *multiple* critics judging a given state $s$. The gradients w.r.t. the critics then become:

$$\nabla_{\theta^{\text{short-term}}}[(R_t^{\text{short-term}} - V(s_t; \theta^{\text{short-term}}))^2] + \nabla_{\theta^{\text{long-term}}}[(R_t^{\text{long-term}} - V^{vwr}(s_t; \theta^{\text{long-term}}))^2] \tag{7}$$

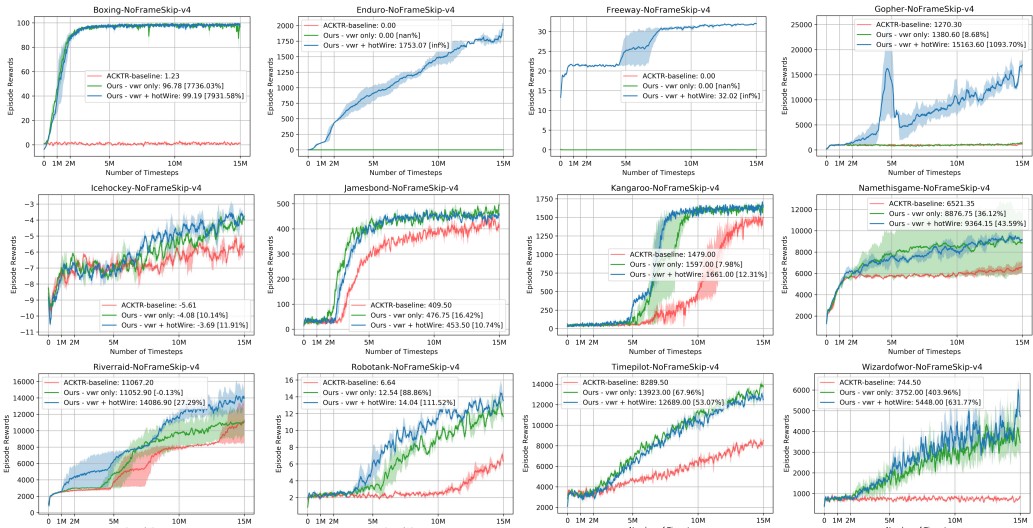

Figure 3: Performance of A2MC on Atari games. "Hot-Wiring" exploration makes the agent easier to figure out how to play challenging games like "Robotank" and "WizardOfWor", and for most games, it provides a better initial state for the agent to start off at a game and hence to obtain better final results. The number in figure legend shows the average reward among the last 100 episodes and the percentage shows the performance margin as compared to ACKTR. The shaded region denotes the standard deviation over 2 random seeds.

where the standard grading clipping approach can be applied in Eq. 7 for enhanced stability. More advanced methods for estimating $R_t^{\text{short-term}}$ and $R_t^{\text{long-term}}$ above, such as the online variant of generalized advantage estimation (GAE) using eligibility traces Schulman et al. (2016) can be adopted in place of Eq. 6, as shown below (see also Appendix G):

$$
\begin{aligned}
A_t^{\text{short-term}} &= \sum_{n=0}^{\infty} (\gamma\lambda)^n \delta_{t+n}^{vwr}, \text{ with } \delta_t = r_t + \gamma V(s_{t+1}) - V(s_t) \\
A_t^{\text{long-term}} &= \sum_{n=0}^{\infty} (\gamma\lambda)^n \delta_{t+n}^{vwr}, \text{ with } \delta_t^{vwr} = r_t^{vwr} + \gamma V^{vwr}(s_{t+1}) - V^{vwr}(s_t)
\end{aligned}
\tag{8}
$$

where the generalized estimator of the advantage function $A_t^{\text{short-term}}$ and $A_t^{\text{long-term}}$ allows a trade-off of bias *v.s.* variance using the parameter $0 \leq \lambda \leq 1$, similar to the TD($\lambda$) approach for eligibility traces. We show the effectiveness of the proposed characterization of past reward statistics in multiple advantage actor-critic frameworks (i.e., ACKTR and PPO), where the two different value functions can share the same low-level feature representation, enabling a single agent to learn multiple critics parameterized by $\theta^j, j \in \{\text{short-term}, \text{long-term}\}$. (See also Appendix I for the full algorithm).

## 5 Hot-Wire $\epsilon$-Exploration

Being handed a game-stick, a human most likely would try out all the available buttons on it to see which particular button entails whatever actions on the game screen, hence receiving useful feedbacks. Inspired by this, we propose to hot-wire the agent to perform an identical sequence of randomly chosen actions in the N-step look-ahead during the initial stage (randomly pressing down a game-stick button for a while):

$$
a_{t+k} = \begin{cases} \text{a random action identical for all k} & \text{with prob } \epsilon \\ \pi(a_{t+k}|s_{t+k}) \text{ for } k = 0, 1, 2, ..., N-1 & \text{with prob } 1-\epsilon \end{cases}
\tag{9}
$$

We show that by enabling the "hot-wiring" mechanism[2], a seemingly trapped agent can be boosted to learn to quickly solve problems where rewards can only be triggered by particular action sequences, as shown in games like "Robotank" and "WizardOfWor" in Figure 3.

---

[2]hot-wire is triggered only when the agent is unable to receive meaningful rewards in an initial learning stage. The legend "vwr + hotWire" in Fig. 3 indicates that the mechanism is "enabled" but not "enforced".

## 6 EXPERIMENTS

We use the same network architecture and natural gradient optimization method as in the ACKTR model Wu et al. (2017). We set $\sigma_{max} = 1.0$, $\tau = 2.0$ and $T = 20$ in the computation of variability-weighted reward (see Appendix C for hyperparameter studies). For hot-wiring exploration, we choose $\epsilon = 0.20$ and initial stage to be first $\frac{1}{40}$ of the total training period for all experiments. Other hyperparameters such as learning rate and gradient clipping remain the same as in the ACKTR model Wu et al. (2017), in addition to adopting GAE Schulman et al. (2016) for a stronger ACKTR baseline (see Sec 4.2). We first present results of evaluating the proposed A2MC model in two standard benchmarks, the discrete Atari experiments and the continuous MuJoCo domain. Then we show ablation studies on the robustness of the hyper-parameters involved as well as evaluating the extensibility of the proposed long/short-term reward characterizations to other on-policy methods. Further extensions to off-policy domains are presented in Appendix E and Appendix F.

### 6.1 ATARI 2600 GAMES

We follow standard evaluation protocol to evaluate A2MC in a variety of Atari game environments (starting with 30 no-op actions). We train our models for 15 million timesteps for each game environment and score each game based on the average episode rewards obtained among the last 100 episodes as in Wu et al. (2017). The learning results on 12 Atari games are shown in Figure 3 where we also included an ablation experiment of A2MC without hot-wiring. We observe that on average A2MC improves upon ACKTR in terms of final performance under the same training budget. Our A2MC is able to consistently improve agent performance based on the proposed characterization of reward statistics, hence the agent is able to get out of local minima in less time (higher sample efficiency) compared to ACKTR. The complete learning results on all games are attached in the Appendix B.

Table 1: Comparison of average episode rewards at the end of 50 million timesteps in Atari experiments. The reward scores and the first episodes reaching human-level performance Mnih et al. (2015) are reported as in Wu et al. (2017). A2MC is able to solve games that are challenging to ACKTR and also retain comparable performance in the rest of games.

| Domain | Human Level | ACKTR | | A2MC | |
|---|---|---|---|---|---|
| | | Rewards | Episode | Rewards | Episode |
| Asteroids | 47388.7 | 34171.0 | N/A | **830232.5** | **11314** |
| Beamrider | 5775.0 | 13581.4 | 3279 | 13564.3 | **3012** |
| Boxing | 12.1 | 1.5 | N/A | **99.1** | **158** |
| Breakout | 31.8 | **735.7** | 4097 | 411.4 | **3664** |
| Double Dunk | -16.4 | -0.5 | 742 | **21.3** | **544** |
| Enduro | 860.5 | 0.0 | N/A | **3492.2** | **730** |
| Freeway | 29.6 | 0.0 | N/A | **32.7** | **1058** |
| Pong | 9.3 | 20.9 | 904 | 19.4 | **804** |
| Q-bert | 13455.0 | 21500.3 | **6422** | 25229.0 | 7259 |
| Robotank | 11.9 | 16.5 | - | **25.7** | 4158 |
| Seaquest | 20182.0 | 1776.0 | N/A | **1798.6** | N/A |
| Space Invaders | 1652.0 | **19723.0** | 14696 | 11774.0 | **11064** |
| Wizard of Wor | 4756.5 | 702 | N/A | **7471.0** | **8119** |

We further expand the training budget and continue learning the games until 50 million timesteps as in Wu et al. (2017). As shown in Table 1, our A2MC model can solve games like "Boxing", "Freeway" and "Enduro" that are challenging for the baseline ACKTR model. For a full picture of model performance in Atari games, A2MC has a human-level performance rate of $74.5\%$ (38 out of 51 games) in the Atari benchmarks, compared to $63.6\%$ reached by ACKTR. Individual game scores for all the Atari games are reported in the Appendix B.

### 6.2 CONTINUOUS CONTROL

For the evaluations on continuous control tasks simulated in MuJoCo environment, we first follow Wu et al. (2017) and tune a different set of hyper-parameters from Atari experiments. Specifically, all

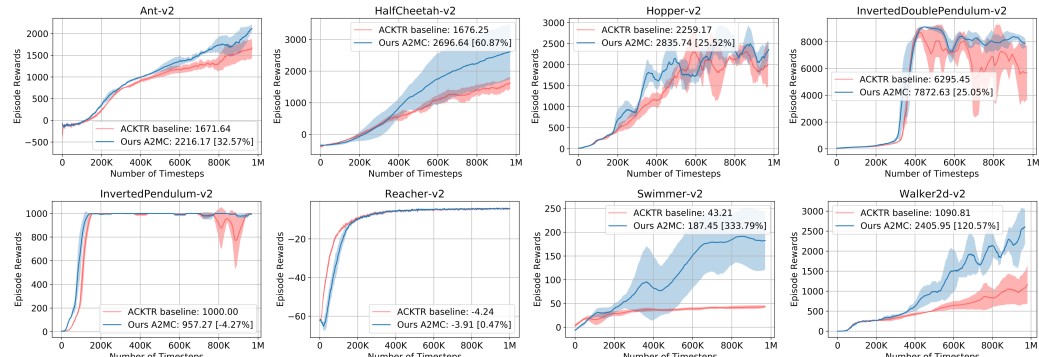

Figure 4: Performance on the MuJoCo benchmark. A2MC is also competitive on MuJoCo continuous domain when compared to ACKTR. The shaded region denotes std over 3 random seeds.

MuJoCo experiments are trained with a larger batch size of 2500. The results of eight MuJoCo environments trained for 1 million timesteps are shown in Figure 4. We observe that A2MC also performs well in all MuJoCo continuous control tasks. In particular, A2MC has brought significant improvement on the tasks of *HalfCheetah*, *Swimmer* and *Walker2d* (see Table 2).

To test the robustness of A2MC, we perform another set of evaluations on MuJoCo tasks by keeping an identical set of hyper-parameters used in the Atari experiments. Figure 7 in Appendix C shows this ablation result. We observe that even under sub-optimal hyper-parameters, our A2MC model can still learn to solve the MuJoCo control tasks in the long run. Moreover, it is less prone to overfitting when compared to ACKTR under such "stress testing". Additional hyper-parameter studies can be found in Appendix C.

We also evaluate a multi-critics variant of the proximal policy optimization (PPO) model on the MuJoCo tasks with our proposed long/short-term rewards. In particular, we observe that our proposed variability-weighted reward generalizes well with the vanilla PPO, and our multi-critics PPO variant (MC-PPO) shows more favorable performance, as shown in Table 2. Specifically, MC-PPO shows the best performance on *Hopper* and *Walker-2d* among all models under the 1-million timesteps training budget. Both of our multi-critics variants (A2MC and MC-PPO) have won 6 out of the 8 MuJoCo tasks with relative performance margins (percentages in parentheses) larger than 25% (see Table 2).

Table 2: Average episode rewards obtained among the last 10 episodes upon 1 million timesteps of training in MuJoCo experiments.

| GAMES | ACKTR | Our A2MC | | PPO | | Our MC-PPO | |
|---|---|---|---|---|---|---|---|
| Ant | 1671.6 | **2216.1** | **(32.5%)** | 411.4 | ($\pm$ 107.7) | **618.9** | **(50.4%)** |
| HalfCheetah | 1676.2 | **2696.6** | **(60.8%)** | 1433.7 | ($\pm$ 83.9) | **2473.4** | **(72.5%)** |
| Hopper | 2259.1 | **2835.7** | **(25.5%)** | 2055.8 | ($\pm$ 274.6) | **3131.3** | **(52.3%)** |
| InvertedDoublePendulum | 6295.4 | **7872.6** | **(25.0%)** | 4454.1 | ($\pm$ 1098.1) | **7648.7** | **(71.7%)** |
| InvertedPendulum | 1000.0 | 957.2 | (-4.2%) | 839.7 | ($\pm$ 127.1) | 777.4 | ($-7.4$%) |
| Reacher | -4.2 | -3.9 | (0.4%) | -5.47 | ($\pm$ 0.3) | $-10.3$ | ($-8.5$%) |
| Swimmer | 43.2 | **187.4** | **(333.7%)** | 79.1 | ($\pm$ 31.2) | **102.9** | **(30.2%)** |
| Walker2d | 1090.8 | **2405.9** | **(120.5%)** | 2300.8 | ($\pm$ 397.6) | **3718.1** | **(61.6%)** |
| Win — Fair — Lose | N/A | 6 — 2 — 0 | | N/A | | 6 — 2 — 0 | |

## 7 CONCLUSION

In this work, we introduce an effective auxiliary reward signal to remedy the deficiencies of learning solely from the standard environment rewards. Our proposed characterization of past reward statistics results in improved learning and higher sample efficiencies for on-policy methods, especially in challenging tasks with sparse rewards. Experiments on both discrete tasks in Atari environment and MuJoCo continuous control tasks validate the effectiveness of utilizing the proposed long/short-term reward statistics for on-policy methods using multi-critic architectures. This suggests that expanding the form of reward feedbacks in reinforcement learning is a promising research direction.

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

APPENDIX

## A  EFFECTS OF FLIPPING

While introducing the variability-weighted reward, a flipping operation is conducted in the pre-processing of the reward sequence as formulated in Eq. (3). In Figure 5 and 6, we construct 4 reward sequences to show that the flipping operation can further penalize the oscillation in the recent past rewards while encourage recent stable rewards. *(a1, a2, b1, b2)* share the same value of immediate reward at $t = 9$ and thus the $\mathcal{R}_H$ of all reward sequences are the same. Therefore,

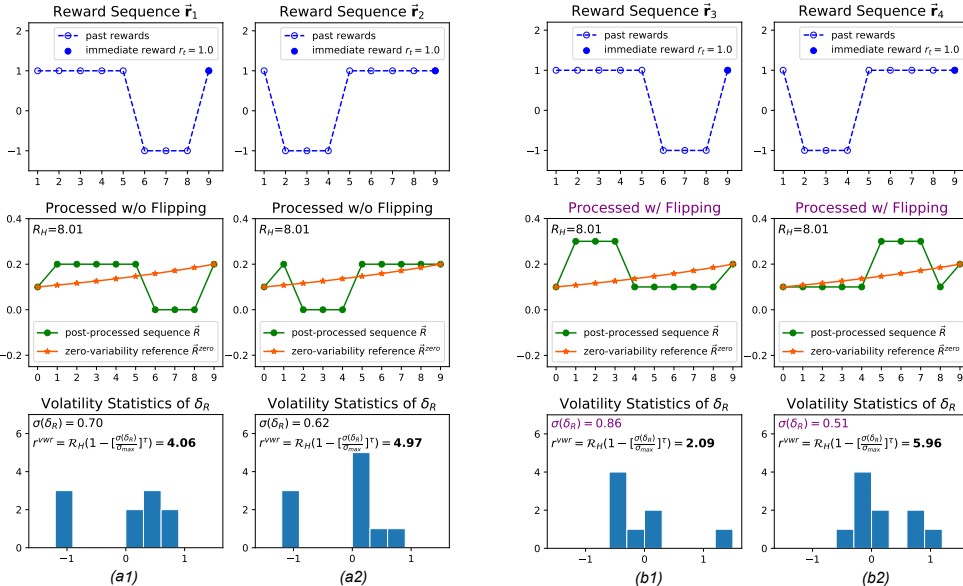

Figure 5: Calculation without flipping.     Figure 6: Calculation with flipping.

the variability-weighted reward only depends on the volatility statistics of $\delta_R$, i.e., how varied past rewards were.

**Without flipping.** In Figure 5, sequence *(a1)* and *(a2)* are mirror symmetrical to the $y$-axis, and the only difference between them is that the recent past rewards ($t = 5, 6, 7, 8$) of *(a2)* are more stable than *(a1)*. Intuitively, we want to encourage stable past rewards like *(a2)* while penalizing oscillation in *(a1)*. As presented in the third row of Figure 5, the $r^{vwr}$ difference of *(a1)* and *(a2)* is less than 1 without flipping in the pre-processing.

**With flipping**. In Figure 6, *(b1, b2)* exactly have the same reward sequence as *(a1, a2)*, respectively. However, flipping is performed as a step of pre-processing, largely increasing the $r^{vwr}$ gap (from less than 1 to nearly 4) between the two constructed sequences. Comparing *(b1, b2)* with *(a1, a2)*, the post-processed sequences $\vec{\mathcal{R}}$ (shown in green) become centrosymmetric to those without flipping. Specifically, the recent reward drops at $t = 6, 7, 8$ are reflected as high values at the beginning of $\vec{\mathcal{R}}$ as shown in *(b1)*, while oscillations long ago are transformed into high values at the end of $\vec{\mathcal{R}}$ as presented in *(b2)*. When compared to the zero-variability reference (shown in orange), which is designed as an exponential function, the flipping leads to a higher variability for the former sequence while a lower variability for the latter one, enlarging the $r^{vwr}$ gap between those two sequences.

## B    COMPLETE RESULTS IN ATARI 2600 GAMES

We show the learning curves for 15 million timesteps on all Atari games in Figure 12 and in Table 3 we show the complete results of training til 50 million timesteps. report the mean episode reward as in Wu et al. (2017). Entries with $\sim$ indicates approximated value as retrieved from learning figures published by Wu et al. (2017). Results from other models are taken from Wu et al. (2017) and Mnih et al. (2015). We show that A2MC has reached a human-level performance rate of $74.5\%$ (38 out of 51 games) as compared to $63.6\%$ reached by ACKTR. The relative performance margin of A2MC as compared to ACKTR is also shown.

## C    HYPER-PARAMETER STUDIES

The proposed variability-weighted reward mechanism considers the volatility of rewards by keeping a $T$-step history of agent's performance. The hyper-parameter $T = 20$ is empirically chosen to be the same as the look-ahead parameter $N$ in standard on-policy methods, so as to keep the same period ($T = N = 20$) in "T-step history" and "N-step forward". And $\sigma_{max} = 1$ is chosen as the maximum of the observed volatility based on statistics in the T history rewards of the ACKTR models. As parameter choices could be vital, we perform an additional ablation study shown below. The result shows that the performance of A2MC is robust across different parameters of choice and is not too sensitive to changes on either of the hyper-params.

| Games | ACKTR | A2MC w/ | $T{=}20$ $\sigma_{max}{=}1$ | $T{=}10$ $\sigma_{max}{=}1$ | $T{=}10$ $\sigma_{max}{=}2$ | $T{=}40$ $\sigma_{max}{=}1$ | $T{=}40$ $\sigma_{max}{=}2$ |
|---|---|---|---|---|---|---|---|
| Boxing | 1.23 | | **99.19** | 94.76 | 98.51 | 99.18 | 98.07 |
| Jamesbond | 409.50 | | 453.50 | 438.50 | **470.00** | 442.25 | 457.75 |
| Wizard of Wor | 744.50 | | 5448.00 | **5601.00** | 5363.50 | 2528.50 | 3287.50 |

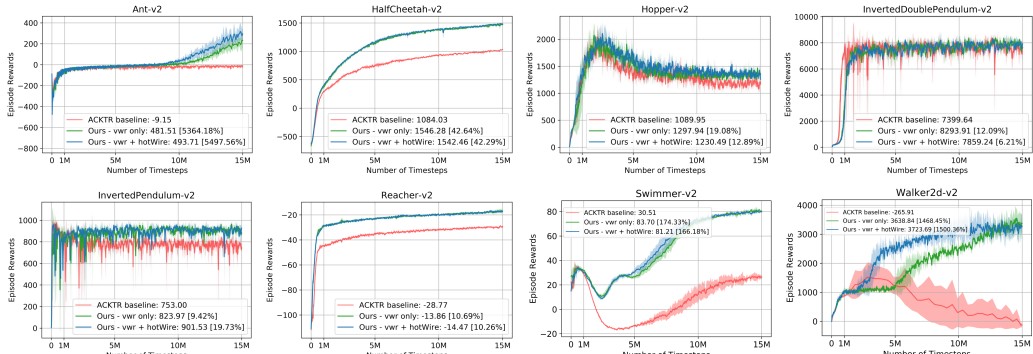

Figure 7: "Stress testing" ablation study on the MuJoCo continuous benchmark using hyperparameters tuned in Atari discrete control. Although this set of hyperparameters is suboptimal for the MuJoCo continuous control tasks, A2MC still obtain reasonable performance in the long run and it is less prone to overfitting.

Table 3: Raw scores across all games, starting with 30 no-op actions. Scores are reported by averaging the last 500 episodes upon 50 million timesteps of training as in Wu et al. (2017). A relative margin comparing A2MC to ACKTR is shown. Scores from other models are taken from Wu et al. (2017) and Mnih et al. (2015).

| GAME | Human | DQN | DDQN | Prior. Duel | ACKTR | Our A2MC | (Margin) |
|---|---|---|---|---|---|---|---|
| Alien | 7127.7 | 1620 | 3747.7 | 3941 | 3197.1 | 2986.3 | -6.6% |
| Amidar | 1719.5 | 978 | 1793.3 | 2296.8 | 1059.4 | 2040.1 | **92.6%** |
| Assault | 742.0 | 4280.4 | 5393.2 | 11477 | 10777.7 | 9892.4 | -8.2% |
| Asterix | 8503.3 | 4359 | 17356.5 | 375080 | 31583.0 | 32671.0 | 3.4% |
| Asteroids | 47388.7 | 1364.5 | 734.7 | 1192.7 | 34171.6 | 828931.6 | **2325.8%** |
| Atlantis | 29028.1 | 279987 | 106056 | 395762 | 3433182.0 | 2886274.0 | -15.9% |
| Bankheist | 753.1 | 455 | 1030.6 | 1503.1 | 1289.7 | 1290.6 | 0.1% |
| Battlezone | 37187.5 | 29900 | 31700 | 35520 | 8910.0 | 10570.0 | **18.6%** |
| Beamrider | 16926.5 | 8627.5 | 13772.8 | 30276.5 | 13581.4 | 13715.6 | 1.0% |
| Berzerk | 2630.4 | 585.6 | 1225.4 | 3409 | 927.2 | 974.0 | 5.0% |
| Bowling | 160.7 | 50.4 | 68.1 | 46.7 | 24.3 | 31.6 | **30.0%** |
| Boxing | 12.1 | 88 | 91.6 | 98.9 | 1.5 | 93.5 | **6344.8%** |
| Breakout | 30.5 | 385.5 | 418.5 | 366 | 735.7 | 420.6 | -42.8% |
| Centipede | 12017.0 | 4657.7 | 5409.4 | 7687.5 | 7125.3 | 12096.5 | **69.8%** |
| Choppercommand | 9882.0 | N/A | N/A | N/A | ~8000 | 12149.0 | ~**42.5%** |
| Crazyclimber | 35829.4 | 110763 | 117282 | 162224 | 150444.0 | 152439.0 | 1.3% |
| Demonattack | 1971.0 | 12149.4 | 58044.2 | 72878.6 | 274176.7 | 361807.1 | **32.0%** |
| Doubledunk | -16.4 | -6.6 | -5.5 | -12.5 | -0.5 | 20.6 | **3907.5%** |
| Enduro | 860.5 | 729 | 1211.8 | 2306.4 | 0.0 | 3550.6 | ∞% |
| Fishingderby | -38.7 | -4.9 | 15.5 | 41.3 | 33.7 | 38.4 | **13.9%** |
| Freeway | 29.6 | 30.8 | 33.3 | 33 | 0.0 | 32.7 | ∞% |
| Frostbite | 4335.0 | N/A | N/A | N/A | ~280 | 293.7 | ~5.1% |
| Gopher | 2412.5 | 8777.4 | 14840.8 | 104368.2 | 47730.8 | 86101.4 | **80.4%** |
| Gravitar | 2672.0 | N/A | N/A | N/A | ~300 | 995.0 | -2.9% |
| Icehockey | 0.9 | -1.9 | -2.7 | -0.4 | -4.2 | -2.1 | **16.3%** |
| Jamesbond | 302.8 | 768.5 | 1358 | 812 | 490.0 | 545.0 | **11.2%** |
| Kangaroo | 3035.0 | 7259 | 12992 | 1792 | 3150.0 | 11269.0 | **257.7%** |
| Krull | 2665.5 | 8422.3 | 7920.5 | 10374.4 | 9686.9 | 10245.4 | 5.8% |
| Kungfumaster | 22736.3 | 26059 | 29710 | 48375 | 34954.0 | 39773.0 | **13.8%** |
| Mspacman | 15693.0 | N/A | N/A | N/A | ~3500 | 5006.1 | ~**34.5%** |
| Namethisgame | 4076.0 | N/A | N/A | N/A | ~12500 | 12569.9 | ~0.6% |
| Phoenix | 7242.6 | 8485.2 | 12252.5 | 70324.3 | 133433.7 | 221288.3 | **65.8%** |
| Pitfall | 6463.7 | -286.1 | -29.9 | 0 | -1.1 | -2.5 | -0.3% |
| Pong | 14.6 | 20.9 | 21 | 20.9 | 20.9 | 19.7 | -5.9% |
| Privateeye | 69571.0 | N/A | N/A | N/A | ~560 | 507.0 | -9.5% |
| Qbert | 13455.0 | 13117.3 | 15088.5 | 18760.3 | 23151.5 | 24075.8 | 4.0% |
| Riverraid | 17118.0 | 7377.6 | 14884.5 | 20607.6 | 17762.8 | 18671.9 | 5.1% |
| Roadrunner | 7845.0 | 39544 | 44127 | 62151 | 53446.0 | 50071.0 | -6.3% |
| Robotank | 11.9 | 63.9 | 65.1 | 27.5 | 16.5 | 26.5 | **60.5%** |
| Seaquest | 42054.7 | 5860.6 | 16452.7 | 931.6 | 1776.0 | 1805.6 | 1.7% |
| Solaris | 12326.7 | 3482.8 | 3067.8 | 133.4 | 2368.6 | 2277.2 | -3.9% |
| Spaceinvaders | 1668.7 | 1692.3 | 2525.5 | 15311.5 | 19723.0 | 13544.2 | -31.3% |
| Stargunner | 10250.0 | 54282 | 60142 | 125117 | 82920.0 | 89616.0 | 8.1% |
| Tennis | -8.9 | N/A | N/A | N/A | ~-12 | -4.7 | ~**20.4%** |
| Timepilot | 5229.2 | 4870 | 8339 | 7553 | 22286.0 | 21992.0 | -1.3% |
| Tutankham | 167.6 | 68.1 | 218.4 | 245.9 | 314.3 | 193.7 | -38.4% |
| Upndown | 11693.2 | 9989.9 | 22972.2 | 33879.1 | 436665.8 | 563659.3 | **29.1%** |
| Videopinball | 17667.9 | 196760.4 | 309941.9 | 479197 | 100496.0 | 127452.4 | **26.8%** |
| Wizardofwor | 4756.5 | 2704 | 7492 | 12352 | 702.0 | 7864.0 | **1020.2%** |
| YarsRevenge | 54576.9 | 18098.9 | 11712.6 | 69618.1 | 125169.0 | 143141.5 | **14.4%** |
| Zaxxon | 9173.3 | 5363 | 10163 | 13886 | 17448.0 | 19365.0 | **11.0%** |
| Human-level (Win / Total) | N/A | 21 / 44 (47.7%) | 31 / 44 (70.4 %) | 34 / 44 (77.3 %) | 28 / 44 (63.6 %) | **38 / 51** (74.5 %) | |

# D    EXTENSION TO MULTI-CRITIC PPO (MC-PPO)

The learning results of the proposed MC-PPO model on the MuJoCo tasks are shown in Figure 8. MC-PPO shows the best performance on *Hopper* and *Walker-2d* among all models under the 1-million timesteps training budget. Both of our multi-critics variants (A2MC and MC-PPO) have won 6 out of the 8 MuJoCo tasks with relative performance margins (percentages in parentheses) larger than 25%.

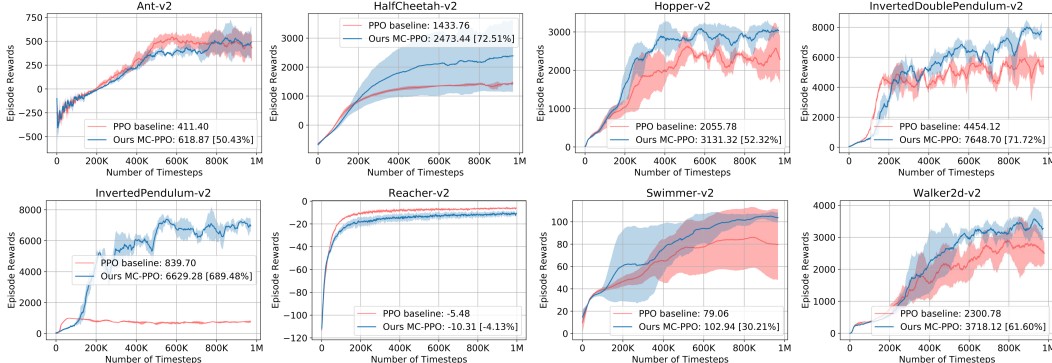

Figure 8: Performance on the MuJoCo continuous control benchmarks using PPO-based methods. Our proposed long/short-term reward characterization can be extended to the PPO method, i.e., the proposed multi-critic variant of PPO (MC-PPO). The shaded region denotes the standard deviation over 3 random seeds.

# E EXTENSION TO OFF-POLICY METHODS

Methods involving experience replay belong to the family of off-policy methods as they learn from off-policy trajectories. They were considered to be beyond the scope of this work, as we set out to improve the family of "on-policy" methods and we try to present as complete the analyses as possible (on both Atari and MuJoCo) in the main text.

Notwithstanding this, we have been actively exploring the potential of applying the proposed reward mechanism with off-policy methods (in particular, on the strong baseline *Rainbow* Hessel et al. (2018). For consistencies in comparisons, all hyperparameters (e.g., learning rate, distributional atoms, noisy net $\sigma_0$) are kept identical as in Hessel et al. (2018) except that we used a smaller replay buffer size of 50,000 for both the baseline and our method (due to limited compute). Moreover, we use the same experiment settings as in Sec 6 and we have *NOT* further tuned any parameters in VWR. We show preliminary results at 10 million time steps on Atari games in Figure 9 and we observe it is promising that introducing the proposed characterization of variability-weighted reward mechanism improves off-policy methods as well.

The robustness of our proposed reward mechanism across both on-policy and off-policy frameworks suggests that the concept of "risk-adjusted return" Sharpe (1994) should apply in reinforcement learning in general, as it brings the desired property in faciliating better sample efficiency and learning stability. Given limited time and computing resources we are not able to present a full analysis on all the off-policy frameworks as we did for the on-policy methods within this paper (since training off-policy models takes significantly longer time). Potentially we aim to have the complete results in an additional paper in our future works.

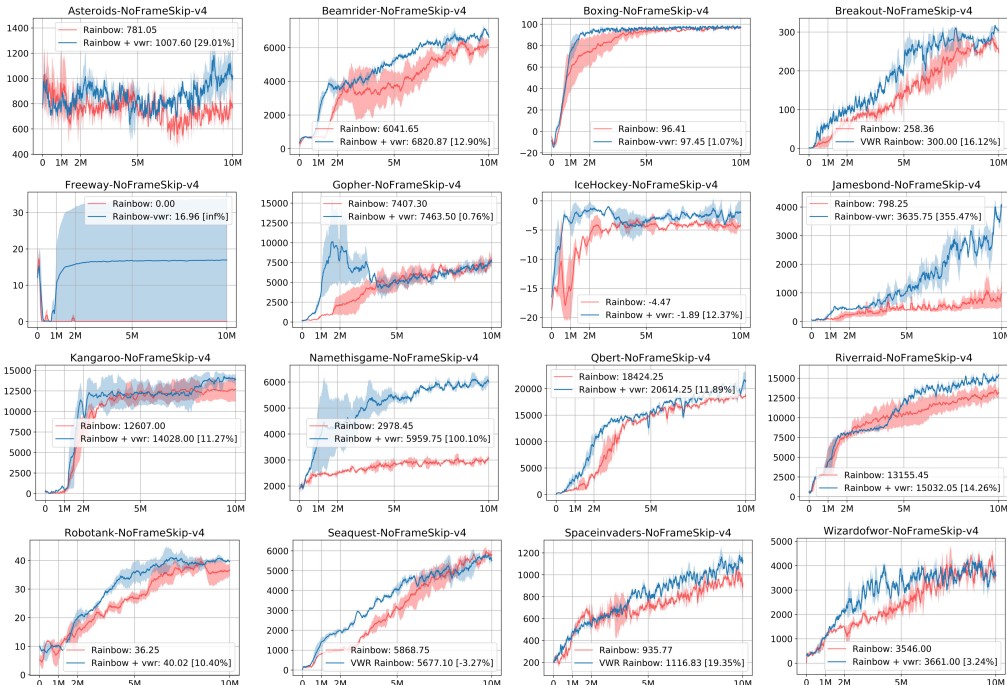

Figure 9: Performance of applying the variability-weighted reward to the *Rainbow* model on the Atari benchmark. We observe that introducing the proposed reward characterizations significantly expedite the learning in games such as "Jamesbond" and "NameThisName", while showing consistent improvement towards the rest. The shaded region denotes the standard deviation over 2 random seeds.

# F CASE STUDY: PLAYING DOOM WITH REWARD SHAPING

It is worth investigating whether the proposed auxiliary reward signal VWR can work "side-by-side" with carefully shaped rewards specific to some particular game scenario – for example, the FPS game *Doom* Lample & Chaplot (2017). As our proposed reward characterization is generic in design and orthogonal to reward shaping, we aim to validate that the concept of risk-adjusted return and variability weights can be equally applied under such shaping settings.

To this end, we adopt the off-policy agent "Arnold" Lample & Chaplot (2017) with experience replay as our baseline and we calculate VWR (see Section 4]) based on the historical sequence of the shaped rewards defined in Lample & Chaplot (2017) (See the Table 4). For VWR parameters, we set $\sigma_{max} = 5$ since the maximum (minimum) attainable reward is $5.0$ $(-5.0)$ under such reward shaping[3]. The rest of the game setup and bot numbers are defaulted to the code released by Lample & Chaplot (2017).

Table 4: Reward shaping settings as in Arnold Lample & Chaplot (2017)

| Type | Base / Dist | Kill | Suicide | Death | Injured | Use ammo | Weapon / Ammo / Medkit /Armor |
|------|-------------|------|---------|-------|---------|----------|-------------------------------|
| Value | 0.0 | 5.0 | -5.0 | -5.0 | -1.0 | -0.2 | 1.0 / 1.0 / 1.0 / 1.0 |

We follow the evaluation criterion of Track-1 in ViZDoom AI Competition 2016 using "Frags per episode", i.e., the number of kills minus the number of suicides for the agent in one round of game (higher is better). The result under 50 training hours is shown in Figure 10 and we consistently observe that the Arnold agent can be significantly boosted with the help of VWR. This confirms that our proposed reward characterization is able to bring further improvements on top of both reward shaping and experience replay methods across domains.

Figure 10: Doom - Limited Deathmatch (Track-1)

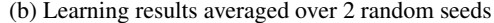

(a) Game statistics in 50 hours      (b) Learning results averaged over 2 random seeds

| After 24 hours | Arnold | Arnod + VWR |
|----------------|--------|-------------|
| Kills | 105 | **183** |
| Frags | 87 | **173** |
| K/D ratio | 1.48 | **2.08** |

| After 50 hours | | |
|----------------|--------|-------------|
| Kills | 116 | **244** |
| Frags | 113 | **223** |
| K/D ratio | 2.00 | **2.65** |

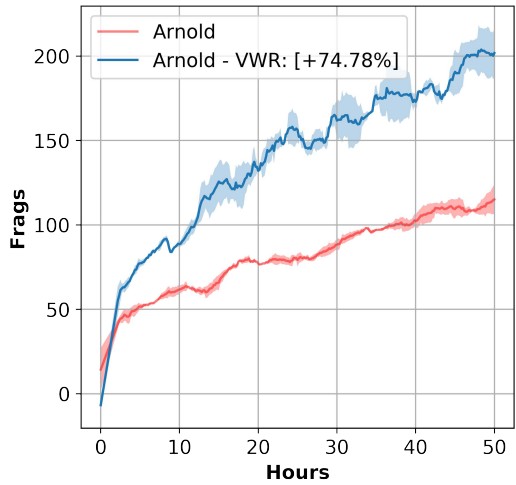

---

[3]The maximum of the volatility based on statistics in historic reward of length $T = 20$ (e.g., in a scenario where the past 20 rewards are $[+5.0] \times 10 + [-5.0] \times 10$) gives a good choice for $\sigma_{max} = 5.0$.

# G   ABLATION STUDY: VWR V.S. ELIGIBILITY TRACE

Eligibility traces TD($\lambda$) is widely used in bridging TD algorithms to Monte Carlo (MC) methods. Essentially, the discounted cumulative return can be formulated by not just toward "any n-step" return (using n-step look ahead), but toward any *average* of n-step look-ahead returns Sutton & Barto (2018). The online variant of generalized advantage estimation using eligibility traces (GAE) Schulman et al. (2016) confirms that on-policy methods can benefit from TD($\lambda$) learning.

For the proposed variability-weighted reward, the design theme is to look explicitly backward and to assess the past performance of the agent via the "risk-adjusted return" concept. These two mechanisms can be combined seamlessly via Eq. 8 and our empirical results suggest VWR brings further improvements on top of eligibility traces.

As VWR and eligibility traces are thematically similar in some sense, we further perform an ablation study to contrast the contributions brought by VWR. As shown in Figure 11, we compare three different settings: (1) ACKTR + GAE, (2) ACKTR + vwr and (3) ACKTR + GAE + vwr (i.e., the proposed A2MC). We observe that on average VWR brings greater improvements compared to eligibility traces, and the combination of both (i.e., A2MC) results in consistently good performance across the Atari testbed.

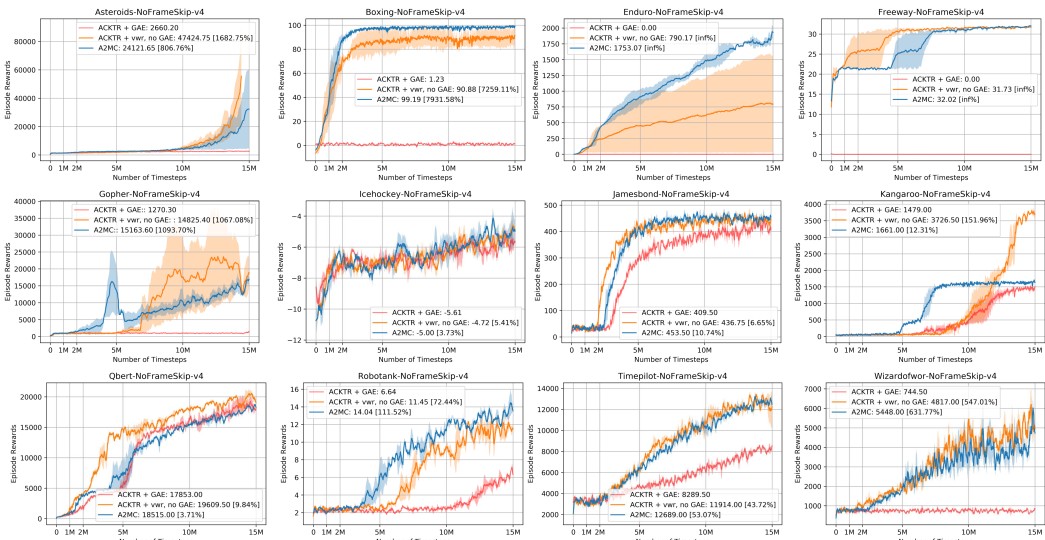

Figure 11: Ablation study of separately applying the (1) the eligibility traces (GAE) and (2) variability-weighted reward (VWR) to the ACKTR model on the Atari benchmark. We observe that the combination of both (i.e., A2MC) results in consistently good performance across the Atari testbed. The shaded region denotes the standard deviation over 2 random seeds.

## H   THE SHARPE RATIO ITSELF

We have explored other forms of reward that fits the general idea of introducing variability weights to the reward shaping mechanism. One example is the "Sharpe Ratio" itself, which is defined as $r^{SR} = \frac{\mathbb{E}[r]}{\sigma(r)}$. In our initial studies, we found it only improved upon the baseline marginally, as $r^{SR}$ could end up emphasizing on penalizing high-variations and it might discourage the agent too intensively (see Figure below). Thats why we have sought an alternative formulation using the proposed $r^{vwr}$ and found that $A2MC_{VWR} > A2MC_{SR} > ACKTR$. An example highlighting the $vwr$ benefit is provided in Appendix A and a more thorough survey on key components in reward designs/formulations will be included in our future works.

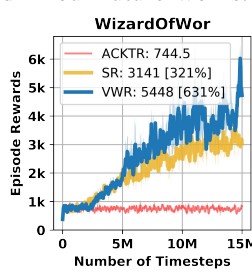

## I   ALGORITHM

The learning algorithm of A2MC is shown in Algorithm 1.

---

**Algorithm 1** Advantage Actor Multi-Critic Learning (A2MC)

---

1: Initialize parameters: $\theta_a, \theta_v^j, j \in \{\text{short-term}, \text{long-term}\}$
2: Initialize look-ahead steps: $N$, step counter: $T = 0$, maximum step: $T_{max}$
3: Initialize hot-wire probability: $\epsilon$
4: Initialize environment: $Env$
5: Initialize reward history: $\vec{r}$
6: **repeat**
7:     Reset gradients: $d\theta \leftarrow 0$ and $d\theta_v^j \leftarrow 0, j \in \{\text{short-term}, \text{long-term}\}$
8:     Get state: $s_t \leftarrow Env$
9:     $flag = 1$, $a_{rand}$ is uniformly sampled in action space with probability $\epsilon$, otherwise $flag = 0$
10:     **for** $t = 0 : N - 1$ **do**
11:         Perform $a_t$ according to policy $\pi(a_t|s_t; \theta_a)$ **if not** $flag$ **else** $a_t = a_{rand}$
12:         Received reward $r_t$ and new state $s_{t+1}$, append $r_t$ to $\vec{r}$
13:         Calculate $r_t^{vwr}$ from $\vec{r}$ based on Eq. (2-7)
14:         $T \leftarrow T + 1$
15:     **end for**
16:     $R^{\text{short-term}} = V(s_N; \theta_v^{\text{short-term}})$
17:     $R^{\text{long-term}} = V(s_N; \theta_v^{\text{long-term}})$
18:     **for** $i = N - 1$ **to** $0$ **step** $-1$ **do**
19:         $R^{\text{short-term}} \leftarrow r_i + \gamma R^{\text{short-term}}$
20:         $R^{\text{long-term}} \leftarrow r_i^{vwr} + \gamma R^{\text{long-term}}$
21:         Advantange gradients wrt $\theta_a : d\theta_a \leftarrow d\theta_a + \nabla_{\theta_a} \log \pi(a_i|s_i; \theta_a) \sum_j (R^j - V(s_i; \theta_v^j))$
22:         **for** $j \in \{\text{short-term}, \text{long-term}\}$ **do**
23:             Accumulate gradients wrt $\theta_v^j : d\theta_v^j \leftarrow d\theta_v^j + \partial(R^j - V(s_i; \theta_v^j))^2 / \partial\theta_v^j$
24:         **end for**
25:     **end for**
26: **until** $T \geq T_{max}$

---

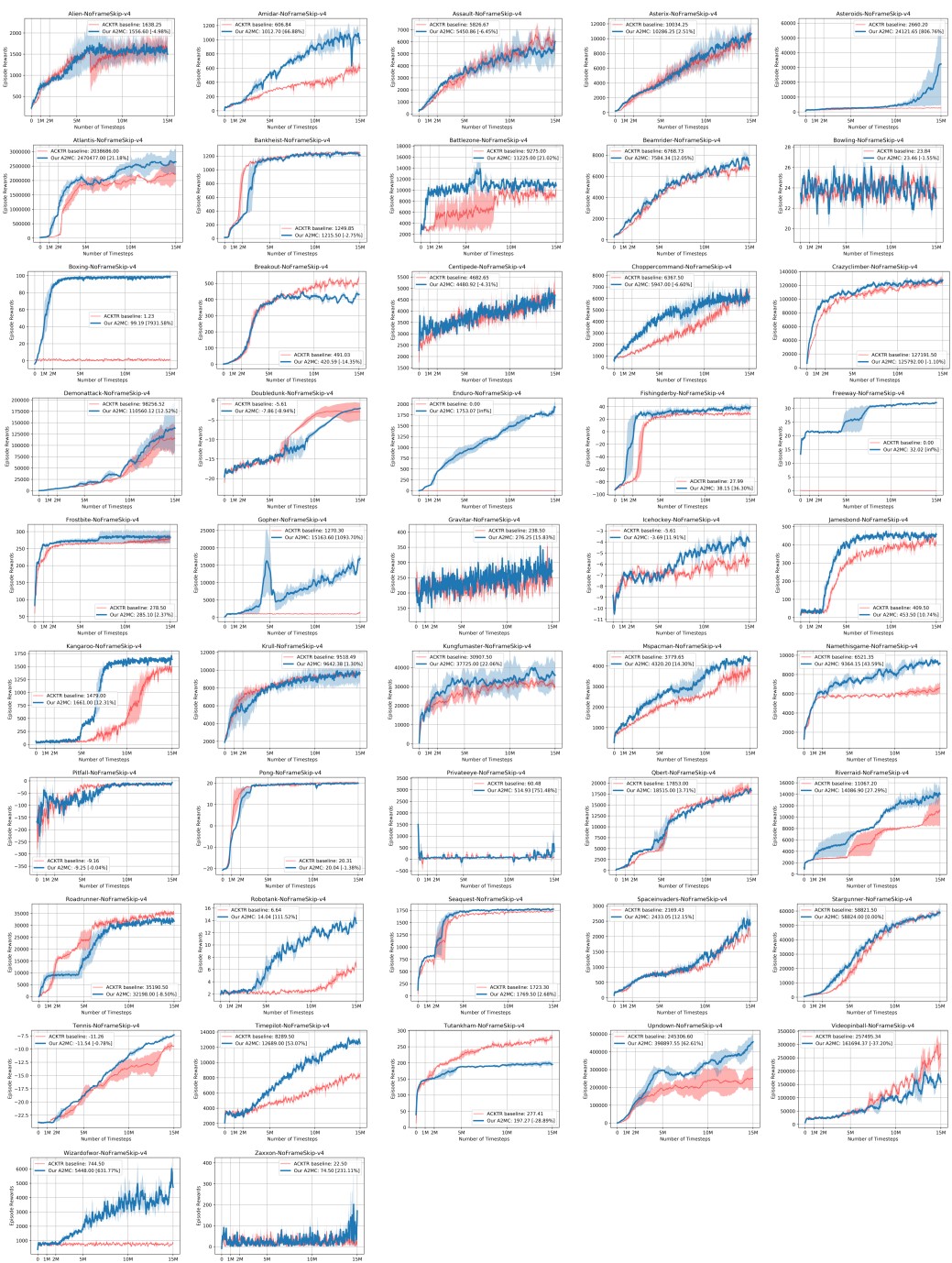

Figure 12: Performance of A2MC on Atari games. The number in figure legend shows the average reward among the last 100 episodes upon 15 million timesteps and the percentage shows the performance margin as compared to ACKTR. The shaded region denotes the standard deviation over 2 random seeds.

