# OpenReview forum: "Improving On-policy Learning with Statistical Reward Accumulation"
_ICLR.cc/2019/Conference_

### Official Review · AnonReviewer1 · 2018-11-02
**Potentially interesting idea but lacking clarity and explanations**

**Rating:** 5
**Confidence:** 3

**Review:**

This paper centers around adding a reward term that, as I understand it, rewards the agent for having seen sequences of rewards that have low variability. This is an interesting idea, however I find the clarity of the main part of the paper (section 4.1, where this new term is defined) quite poor. That section makes several seemingly arbitrary choices that are not properly explained, which makes one wonder if those choices were made mostly to make the empirical results look good or if there are some more fundamental and general concepts being captured there. In particular, I have to wonder where the 100 in the definition of R_H comes from, and also how sigma_max would be determined (it is very hard to get a good intuition on such quantities as an RL practitioner).

The paper also introduces “hot-wire exploration”, basically trying the same action for a while during the initial stage, which is a nice exploration heuristic for Atari, but I am not sure how generally applicable the idea is beyond the Atari testbed.

In general, I am always a bit wary of experimental results that were obtained as a result of introducing additional hyper-parameters or functional forms. However, the results look pretty good, and the authors do manage to show some amount of hyperparameter robustness, which makes me wish the design choices had been more clearly explained..

---

> ### Author Response · Authors · 2018-11-06
> **Explanation of design choices**
>
> Thank you for your kind suggestions and indeed we should have made section 4.1 clearer in the first place.
>
> Q1. [Design choices]
> Ans.
> i. Regarding the multiplying factor 100 in R_H, we should have further explained that the term inside the parenthesis is actually:
>
>     exp( 1/T ln(R_T / R_0) )  - 1
> = (R_T)^(1/T) / (R_0)^(1/T) - 1
> = ((R_T)^(1/T) - (R_0)^(1/T)) / ((R_0)^(1/T))
>
> in the form of (B - A) / A.
> Namely, it describes how large (in the relative sense) the return at the immediate step T is when compared to the initial step R_0, “averaged” by the exponent 1/T over the sequence. By multiplying 100 we were essentially treating it as a relative “percentage” to make it numerically stable across the training process. We also explained why we did "flipping" in Appendix A as we ran out of space in the main text.
>
> ii. And sigma_max = 1 is chosen as the maximum of the observed volatility based on statistics in the T history rewards of the ACKTR models, as we hypothesized that too volatile rewards might impede the learning process of on-policy methods. As shown in Figure 2, we performed statistical analyses on the reward sequence. For instance, reward sequences in Figure 2(a) and Figure 2(b) are relatively more volatile, and their sigma values are around 0.6. Reward sequences with significantly higher volatility may suggest the agent is not performing optimally. So it also makes intuitive sense that a sigma_max=1 as the upper bound should be a good estimate of the maximum tolerated volatility for the reward design.
>
> Nonetheless, we found that the sigma_max choice was not too sensitive, as supported by our ablation study in Appendix C. Moreover, we also showed that the same hyper-parameter setting is equally applicable to MuJoCo tasks (we did not further tune any parameters in the VWR), so we believe this idea generally works for on-policy methods. This was further supported by extending such reward mechanisms to the PPO model in Table 2.
>
> To briefly conclude, the experimental results resonate with the idea that in general, aided by a smooth and less volatile reward design as an auxiliary feedback, on-policy learning can be improved towards better performance. We hope that Figure 2 and Appendix Figure 6 and Figure 7 help to give a good illustration of the reward design and we will try to add to our text as many explanatory details as possible.
>
> Q2. [Hot-wire heuristic]
> Ans.
> i. We admit that this heuristic comes from the Atari testbed where we observe that on some games, off-policy methods can solve them easily while on-policy approaches got stuck from the beginning till end. And as hinted in the main text, hot-wire is designed to “boost” a seemly trapped agent in the initial stage of the games. The design takeaway is, we drew intuition from “epsilon exploration” in off-policy frameworks where the experience replay buffer is initially filled with transitions resulted from random actions; in the on-policy counterpart, as the agent learns with a sequence of actions per update, we decided to try some randomly chosen identical actions for a while if needed in the initial stage.
>
> ii. And as mentioned in our main text, we didn’t enforce hot-wire to be performed for all the Atari games; it depends on whether the task at hand (the reward emitting mechanism of the environment) would require some particular action sequence to be triggered before giving out the first meaningful positive reward as the feedback signal. And outside of the Atari testbed, we indeed observed that hot-wire facilitated learning on "MuJoCo Ant" and "MuJoCo Walker2d", as shown in Figure 5. For another example of using hot-wire in some other domains, we were working on learning a trading agent that may have to perform a sequence to “buy” actions to incur inventory before some appropriate “sell” actions can result in positive rewards. And we did find that hot-wire exploration helps the trading agent quickly get on the right track.

---

### Official Review · AnonReviewer4 · 2018-11-06
**lacks principled derivation but good empirical results**

**Rating:** 4
**Confidence:** 3

**Review:**

Recommendation: Weak reject

Summary:
The paper proposes a variant of deep reinforcement learning (A2MC) for environments with sparse rewards.  The approach replaces the standard environment reward function with a combination of the current reward and the variability of rewards in the last T timesteps, with a goal of decreasing variability.  The authors further propose a “hot-wiring” exploration strategy to bootstrap agents by taking either random actions or actions that have been done in the recent history.  Empirical evaluations in standard benchmarks including several sparse reward Atari games show empirical improvement of this approach over a baseline (ACKTR).


Review:

The paper has strong empirical results that show the A2MC outperforming or reaching the same performance as the baselines in a large number of Atari and MuJoCo domains.  The authors also provide results with and without the hot-wiring feature, which helps isolate its contribution.  However, overall the paper lacks theoretical rigor and most of the proposed changes are done without principled reasons or convergence guarantees.   There is no way of telling from the current paper whether these changes could lead to divergence or suboptimal behavior in other domains.  Examples of such changes include:

* The averaging of the reward terms at different timescales in Equation 4 is the core of the algorithm but is derived ad-hoc.  Why is this a good equation?  Is lack of variability really a desired property and may it lead to a suboptimal policy?  Can anything be said about how it changes behavior in a tabular representation?

* The exponential equation with a constant of 100 appears out of nowhere in equation 5.  Is this a general equation that will really work in different domains and reward scales?

* The variability weights in equations 6 and 7 are never tested empirically – what happens if they are left out?  Where did this equation come from?

* Overall, it is unclear if the combination of rewards at different time scales in equation 10 is stable and leads to convergence.  The terms show resemblance to the eligibility trace equations but lack their theoretical properties.

To make the paper ready for publication, the authors need to justify which of these changes are “safe” in that they guarantee the behavior of the algorithm cannot become much worse, or need to point directly to other methods in the literature that have used such changes and cite the pros and cons that were seen with those changes.

Related to the theme above, the paper does not properly cite other methods used with sparse rewards in traditional RL or Deep RL, especially eligibility traces, which seem highly related to the current approach.  The following related work edits are needed:

* The overall approach is thematically similar to eligibility traces (see the standard Sutton and Barto textbook), except that the authors here use variability rather than combining the reward terms directly.  Eligibility traces are built exactly for these sorts of sparse reward problems and combine short and long-term rewards in a TD update. But there has been substantial investigation of their theoretical properties in the tabular and function approximation cases. The current method needs to compare and contrast to this long-standing method both theoretically and empirically.

* Two other methods that should have been considered in the experiments are experience replay and reward shaping, both of which are beneficial with deep RL in sparse domains.  Experience replay is mentioned in the paper but not implemented as a competitor.  I realize ER is not as computationally efficient as the new approach but it is an important (and stable) baseline.  Reward shaping is not mentioned at all, but is again an important and stable baseline that has been used in such problems – see “Playing FPS Games with Deep Reinforcement Learning” (AAAI, 2017).

* Finally, the related work section mentions a lot of competitive algorithms but does not implement any of them in comparison, which makes it hard to claim the current approach is the best yet proposed.

---

> ### Author Response · Authors · 2018-11-07
> **[Part 1 / 3] The design choices capture the general theme of "risk-adjusted return", the idea of which mainly came from the economics and finance literatures**
>
> [Part 1 / 3]
>
> Thank you very much for your detailed comments and we try to address the related concerns as follows.
>
> Q1 [ Is variability a desired property & Eq. 4 … ]
> Ans.
> i. As mentioned in the last paragraph of Page 4, “Under such processing …”, this is to obtain the post-processed reward sequence R (Figure 2, Green Curve). The averaging term 1/(T+1) in Eq. 4 is used to mitigate the effect of appending f_0 = 1, which was added to prevent the numerical instability when all rewards in the sequence are zero.
>
> ii. The idea of adjusting reward for its variability comes mainly from the *Sharpe Ratio* in the economics and finance literature, where it concerns both “the expected differential return” and “the unit of risk associated with the differential return” (Chapter 3 in [1]). What we set out to do is not to completely hope for “Lack of variability” (or else we would have disposed of the original reward term altogether); rather, we seek the regularization effect of introducing this auxiliary reward term so as to facilitate a more *stable* learning process. In this sense, introducing our variability-adjusted term is indeed a desired property. Though we are in no position to claim that the current VWR has the optimal mathematical formulation of all, our strong empirical evidence indeed shows that such an auxiliary reward feedback has indeed *improved* upon the baseline.
>
> iii. Consider a toy example (similar to a multi-arm bandit), where a skilled gambler initially has 10 dollars and he can play on 3 tables, trying to reach the goal of getting 100 dollars:
>
>         Playing on table 1: 50% chance of winning 8 dollars and 50% chance of losing 4 dollars      -- E[R_1] = 2
>         Playing on table 2: 50% chance of winning 4 dollars and 50% chance of winning 0 dollars  -- E[R_2] = 2
>         Playing on table 3: 50% chance of winning 0 dollars and 50% chance of losing 4 dollars      -- E[R_3] < 0
>
> It is not difficult to see that, E[R_1] = E[R_2] > E[R_3], so an agent without considering the *variability of return* may eventually decide to play on table-1 even though it might lead to *gambler’s ruin* with a non-zero probability; or it might take the agent a large number of trials to figure out table-2 is a better choice than table-1. On the contrary, the agent can easily find out that table-2 is the optimal solution if we have considered the variability (risk) of the rewards.
> In this sense, the general idea of introducing the *adjustment-for-risk* concept via variability of return can indeed change the agent’s behavior in this toy example.
>
> [1] “Adjusting for risk: An improved sharpe ratio”, Kevin Dowd. International review of economics &
> Finance, 2000.
>
> Q2. [The multiplying factor 100]
> Ans.
> Regarding the multiplying factor 100 in R_H, we should have further explained that the term inside the parenthesis is actually:
>     exp( 1/T ln(R_T / R_0) )  - 1
> = (R_T)^(1/T) / (R_0)^(1/T) - 1
> = ((R_T)^(1/T) - (R_0)^(1/T)) / ((R_0)^(1/T))
> in the form of (B - A) / A.
> Namely, it describes how large (in the relative sense) the return at the immediate step T is when compared to the initial step R_0, “averaged” by the exponent 1/T over the sequence. By multiplying 100 we were essentially treating it as a relative “percentage” to make it numerically stable across the training process. We also explained why we did "flipping" in Appendix A as we ran out of space in the main text.
>
> Q3 [ Eq.s 6 and 7 – where is variability weights coming from].
> Ans.
> i. Variability weights came from the original formulation of Sharpe Ratio: E[r] / std(r) = E[r] * 1/std(r), where we replace the term 1/std(r) with variability weights (Eq. 6) to introduce a maximum tolerated volatility term (sigma_max).
>
> ii. Simply following Sharpe Ratio formulation is also viable in our preliminary studies but we found that capping this auxiliary reward term with a maximum tolerated volatility leads to better results. We will include this comparison in the appendix in our revised version. As explained in the answer to Q1, introducing the variability concept is to take into consideration the risk-adjusted return idea and this facilitates learning in our empirical results.
>
> Q4. [“Safeness” of Eq. 10 - Rewards at different time scales]
> Ans.
> We follow standard procedures in on-policy frameworks (e.g., A2C, PPO) to perform gradient clippings and have ensured that they are in the same scale. (This is also typically a practice in multi-task learning frameworks and in frameworks involving multiple loss functions that we have seen). And our empirical results suggest that the formulation is indeed stable across different domains (both Atari and MuJoCo).
>
> (see part 2 and part 3 due to character limits)

---

> > ### Author Response · Authors · 2018-11-07
> > **[Part 2 / 3]**
> >
> > [Part 2 / 3]
> >
> > Q5 [Regarding eligibility traces]
> > Ans.
> > We want to clarify that eligibility trace is orthogonal to our work here.
> >
> > “Eligibility traces” was initially used in Q-learning methods and it was later proposed for actor-critic methods in [5]. John Schulman further proposed an improved scheme based on the idea of eligibility trace called *Generalized Advantage Estimation (GAE)* [6], which is being adopted in many on-policy RL-frameworks with advantage actor-critic methods that follow, such as in the latest implementations of A2C/A3C, ACKTR and PPO. Note that both eligibility traces and generalized advantage estimation (see [6]) address the problem of trying to better estimate the *expected total reward* (or expected returns).
> >
> > The notion of “combining the reward terms directly” in eligibility traces/GAE methods is to have an accurate estimate of the expected returns, by looking “forward” into the future; whereas our contribution is to propose the concept of volatility (standard deviation) of returns, essentially introducing the *expected total risk-adjusted reward*, by looking explicitly “backward” into the past. This is not mutually exclusive with the eligibility trace/GAE idea.
> >
> > In fact, in the ACTKR and in our proposed A2MC, the GAE approach were already adopted when we estimate the *expected total reward* w.r.t. “standard rewards r”. And under similar formulation, we have used the GAE approach to estimate the *expected total risk-adjusted reward* (w.r.t. our VWR terms r_vwr). We should have explicitly stated in the paper that the GAE approach has been used instead of simply showing Equation 8 and 9, since it was implemented in both the ACKTR baseline and in A2MC. And since it is the case that our VWR has brought improvements on top of GAE, we believe our proposed variability formulation works well with eligibility traces and they are not contending.
> >
> > We will revise Section 4.2 accordingly to avoid potential confusions.
> >
> > [5] “An analysis of actor/critic algorithms using eligibility traces: Reinforcement learning with imperfect value function”. Kimura, Hajime and Kobayashi, Shigenobu, In ICML, 1998.
> > [6] “High-dimensional continuous control using generalized advantage estimation”, Schulman, John, et al. In ICLR 2016.
> >
> > Q6. [Regarding Reward Shaping]
> > Ans.
> > We should have explicitly mentioned reward shaping as it is indeed related to our work. To our understanding, reward shaping is tweaking the rewards in a way that the agent would learn faster, which is typically carried out by disposing of the original reward and *replacing* it with the reshaped reward, as in the HRA approach [7] cited in our related work section.
> >
> > We do realize the effectiveness of reward shaping, as evidenced in [8]. But there are two reasons why we have not adopted such approaches or included them as baselines:
> >
> >     i. Firstly, crafting such rewards for each game and for tasks in different domains is likely to be difficult and time consuming, and there does not exist a universal setting in the literature that would work for all Atari games or for all MuJoCo tasks. It’ll then be impossible to compare with previous methods if we tailor to each game a specifically-shaped reward scheme, which is like tuning a different set of learning rate, network size, look-ahead steps separately for each of the Atari games.
> >
> >     ii. Secondly, we aim at proposing a *generic* auxiliary reward signal that can work side-by-side with the original reward instead of replacing it or changing the optimality of the problem. And as mentioned in our related work section, we try to do this “without the need to engineer a decomposition of problem-specific environment rewards”. So it would be unfair to directly compare our method with HRA [7] on the game “Atari MsPacman” as HRA was designed specifically on the “Atari MsPacman” game. And there doesn’t seem to exist a generic way to generalize the game-specific reward design idea for “Atari MsPacman” in [7] to the other Atari games (e.g., Enduro) or MuJoCo tasks (e.g.,  Swimmer). Similar arguments apply to why we have not compared against [8] at the time of submission. We will cite those papers in the related work section and try to better explain our rationale in our choices of baselines for comparison.
> >
> > [7] “Hybrid reward architecture for reinforcement learning”, Harm Van Seijen et al., In NIPS, 2017.
> > [8] “Playing FPS Games with Deep Reinforcement Learning”, Guillaume Lample and Devendra Singh Chaplot, In AAAI 2017.
> >
> > (see part 3 due to character limits)

---

> > > ### Author Response · Authors · 2018-11-07
> > > **[Part 3 / 3]**
> > >
> > > [Part 3 / 3]
> > >
> > > Q7. [Regarding Experience replay and selected methods used in comparison]
> > > Ans.
> > > Methods involving experience replay belong to the family of off-policy methods and they were considered to be beyond the scope of the work, as we set out to improve the family of *on-policy* methods at the time of submission, this is why we have opted for the best *on-policy* models (ACKTR for Atari and PPO for MuJoCo) as our baseline at the time of submission. Rather than trying to prove state-of-the-art in everything, we have managed to show that the auxiliary variability reward indeed improves on-policy learning, as this is the main theme of this paper -- improving *on-policy* learning.
> > >
> > > Notwithstanding this, we have been actively exploring applying the proposed reward mechanism with off-policy methods (in particular, a strong baseline Rainbow[9]) and we show preliminary results at 10 million time steps below (similar to Table 1):
> > >
> > > Final Rewards          |  Beamrider  | Jamesbond   | Qbert
> > > Rainbow baseline    |  5508            | 1114               | 18350
> > > Rainbow + vwr         |  6928             | 3887              | 21527
> > >
> > > The numbers above indeed suggest that it is promising that our proposed reward mechanism would improve off-policy methods as well. Given limited rebuttal time and computing resources, we will try our best to include as many experiments as possible by the end of the rebuttal period to show results of generalizing this VWR reward term in off-policy methods. Potentially we aim to have the complete results in an additional paper.
> > >
> > > [9] "Rainbow: Combining improvements in deep reinforcement learning", Hessel, Matteo, et al., In AAAI 2018
> > >
> > >
> > > ---- A side note: Literature of using the risk-adjusted concept in reinforcement learning ----
> > > In earlier literature, there were indeed successful attempts to adopt the risk-adjusted concepts (Sharpe ratio) directly as the reward function in Q-learning, such as [2]. Although most existing works that touched upon such concepts were from the economics and finance literature (e.g., [2][3][4]) and they were designed specifically towards financial/trading applications, they indeed have shown that using risk-adjusted reward (Sharpe ratio) would be better than using the vanilla rewards in training an RL agent [2] (at least for their trading/finance tasks in consideration).
> > >
> > > We believe this concept has the potential to also benefit reinforcement learning in general so we present our attempt in this paper to incorporate variability to improve *on-policy* learning at the time of submission. (And we are actively generalizing this idea to off-policy models in our concurrent works).
> > >
> > > [2] "An algorithm for trading and portfolio management using Q-learning and sharpe ratio maximization", Gao, Xiu, and Laiwan Chan. In ICNIP, 2000.
> > > [3] "Learning to trade via direct reinforcement", Moody, John, and Matthew Saffell. IEEE transactions on neural Networks, 2001.
> > > [4] "An automated FX trading system using adaptive reinforcement learning", Dempster, Michael AH, and Vasco Leemans. Expert Systems with Applications, 2006.
> > >
> > >
> > > We sincerely appreciate that you have read all our responses and we hope that our work can be further improved in this review process. Thank you very much for your consideration.

---

### Author Response · Authors · 2018-11-24
**Summary of paper revisions**

We want to thank the reviewers for their kind and helpful feedback. We've followed all reviewers' comments and addressed the related concerns in our revision, outlined as follows:

1) We revised *Related Work* to clarify the scope of our paper while explicitly mentioning reward shaping;
2) We revised *Section 4.1* to better explain the principles behind the derivation of VWR;
3) We revised *Section 4.2* and *Section 6* to clarify our experimental settings with GAE/Eligibility Traces;

4) We added extensive experiments on experience-replay methods in Appendix-E (on the strong off-policy baseline Rainbow + VWR)
5) We added a case-study on playing doom with reward shaping in Appendix-F (experience replay + reward shaping + VWR)
6) We added an ablation study explicitly for VWR v.s. Eligibility Trace in Appendix-G.

We hope the reviewers can kindly reconsider our paper for publication after revision.

---

### Meta-Review · Area_Chair1 · 2018-12-14
**Nice empirical results, but ad hoc approach**

**Confidence:** 5
**Recommendation:** Reject

**Metareview:**

The paper proposes an interesting idea for efficient exploration of on-policy learning in sparse reward RL problems.  The empirical results are promising, which is the main strength of the paper.  On the other hand, reviewers generally feel that the proposed algorithm is rather ad hoc, sometimes with not-so-transparent algorithmic choices.  As a result, it is really unclear whether the idea works only on the test problems, or applies to a broader set of problems.  The author responses and new results are helpful and appreciated by all reviewers, but do not change the reviewers' concerns.